# The Evolving Concept of Complete Resection in Lung Cancer Surgery

**DOI:** 10.3390/cancers13112583

**Published:** 2021-05-25

**Authors:** Ramón Rami-Porta

**Affiliations:** 1Department of Thoracic Surgery, Hospital Universitari Mútua Terrassa, University of Barcelona, Plaza Dr. Robert 5, 08221 Terrassa, Spain; rramip@yahoo.es; 2Network of Centers for Biomedical Research in Respiratory Diseases (CIBERES) Lung Cancer Group, 08221 Terrassa, Spain

**Keywords:** complete resection, incomplete resection, lung cancer, lung cancer staging, lung cancer surgery, uncertain resection

## Abstract

**Simple Summary:**

In the surgical treatment of lung cancer, the complete removal of the portion of the lung where the cancer is and of the involved adjacent structures is of paramount importance to achieve long-term survival. The International Association for the Study of Lung Cancer (IASLC) proposed a definition of complete resection that included a well-defined type of removal of the regional lymph nodes as a fundamental step. The lymph nodes may contain cancer cells and, if left behind, cancer will soon progress. The IASLC also defined incomplete resection when there is any evidence of persistent cancer after the operation. It also defined an intermediate condition, uncertain resection, when no evidence of residual disease can be proved, but all the conditions of complete resection are not fulfilled. Four validations of the definitions have proved their prognostic value and, therefore, the definitions should be followed when a surgical resection of lung cancer is planned.

**Abstract:**

Different definitions of complete resection were formulated to complement the residual tumor (R) descriptor proposed by the American Joint Committee on Cancer in 1977. The definitions went beyond resection margins to include the status of the visceral pleura, the most distant nodes and the nodal capsule and the performance of a complete mediastinal lymphadenectomy. In 2005, the International Association for the Study of Lung Cancer (IASLC) proposed definitions for complete, incomplete and uncertain resections for international implementation. Central to the IASLC definition of complete resection is an adequate nodal evaluation either by systematic nodal dissection or lobe-specific systematic nodal dissection, as well as the integrity of the highest mediastinal node, the nodal capsule and the resection margins. When there is evidence of cancer remaining after treatment, the resection is incomplete, and when all margins are free of tumor, but the conditions for complete resection are not fulfilled, the resection is defined as uncertain. The prognostic relevance of the definitions has been validated by four studies. The definitions can be improved in the future by considering the cells spread through air spaces, the residual tumor cells, DNA or RNA in the blood, and the determination of the adequate margins and lymphadenectomy in sublobar resections.

## 1. Introduction

Evarts A. Graham performed the first successful pneumonectomy for lung cancer in 1933. Pneumonectomy was not new. It had been performed for treatment of infectious diseases but was associated with a high mortality rate. Graham’s operation was indicated for a squamous cell carcinoma of the left upper lobe involving the lower lobe bronchus that today would be classified as pathologic (p)T2aN1M0. Once the pneumonectomy was completed, radon seeds were inserted into the stump as a sort of adjuvant brachytherapy. A thoracoplasty was subsequently performed to reduce the chest cavity. The patient survived the operation, but there were postoperative events that required reinterventions for removal of pus and for extending the initial thoracoplasty. The patient was discharged 75 days after the operation. An obstetrician–gynecologist himself, he still practiced for 24 years and survived for 6 more, with no evidence of recurrence, outliving his surgeon, who, ironically, died from lung cancer [1,2]. With that operation, Graham achieved the most important objective of lung cancer resection: to prolong the patient’s life with acceptable quality, which was possible because, knowingly or not, Graham had performed a complete resection, the only surgical procedure consistently and reproducibly associated with prolonged survival. 

The benefit of that first pneumonectomy established the removal of the whole lung as the standard operation for the surgical treatment of lung cancer. In fact, in 1951, when Cahan et al. systematized lung resection with lymphadenectomy, they described the removal of the whole lung, which they called radical pneumonectomy, to differentiate it from simple pneumonectomy, that is, without lymphadenectomy [3]. Interestingly, nine years later, in 1960, Cahan described radical lobectomy—the removal of a lobe with specific lymphadenectomy depending on the lobar location of the primary tumor—and stated that radical lobectomy should be reserved for those patients who could not undergo radical pneumonectomy [4]. It was not considered an oncologically sound operation at that time, although by the end of the decade, radical lobectomy had gained credit as a consolidated operation for lung cancer [5]. 

Central to radical pneumonectomy and radical lobectomy was the lymphadenectomy that accompanied the removal of the lung or lobe. Cahan et al. established it as one of the principles of the surgical treatment of lung cancer, among the absence of distant metastases, a reasonable low morbidity and mortality and the absence of other effective therapy. Knowing from experience that the lymph nodes could be involved at the time of resection or in the future, they wrote the following: “If these lymphatic areas are not resected at the time of the original procedure, they would be hidden within the body so that their future involvement by metastases would escape early detection” [3]. Since then, an adequate intrathoracic lymph node assessment has been an important element in all definitions of complete resection.

The importance of nodal involvement in malignant tumors was highlighted by the fact that it was included in the classification of anatomic extent of the tumor as one of its components. Proposed by Pierre Denoix in a series of articles published in the mid-20th century, the tumor, node and metastasis (TNM) classification was adopted by the Union for International Cancer Control (UICC) in 1960, and from 1960 to 1967 the UICC published brochures with the TNM classifications of individual cancers, the one for the lung appearing in 1966 [6]. Since then, it is well known that, in the absence of distant metastases, nodal involvement leads prognosis. 

The mapping of the pulmonary and mediastinal lymph nodes, first published by Tsuguo Naruke in 1967 in Japanese [7] and subsequently in English [8,9] was instrumental for the understanding of the prognostic impact of nodal disease. This map was the forerunner of all the other maps proposed to date by the American Thoracic Society [10], by Mountain and Dresler [11] and by the International Association for the Study of Lung Cancer (IASLC) [12]. 

Starting from the historical background described above, this perspective article will review the evolution of the concept of complete resection in the surgical treatment of lung cancer. 

## 2. The First Attempt to Code Completeness of Resection

The American Joint Committee on Cancer (AJCC) proposed the residual tumor (R) classification and included it in the first edition of its staging manual in 1977 [13]. It was also adopted by the UICC and appeared in the fourth edition of its manual [14]. Table 1 shows the R categories and their descriptors [15]. 

Although generally applied to surgical procedures, the R classification can be used after the administration of any therapy, either alone or in combination, and can be applied to the primary tumor, to the regional lymph nodes and to the distant metastases. Following non-surgical treatments, the R classification is assessed by clinical methods. After resection, it requires the close collaboration between the surgeon and the pathologist in charge of the specimen [15]. The residual tumor classification reflects the effects of therapy, is the base on which to indicate additional treatments and predicts prognosis [16]. The latter was clearly proved in the study of Edwards et al. [17], based on the IASLC database used to inform the eighth edition of the TNM classification for lung cancer [18]. In this study, the 5-year survival rates for patients with R0, R1 and R2 resections were 73%, 36% and 28%, respectively (*p* < 0.0001). There were no statistically significant differences between survival rates of R1 and R2 (*p* < 0.27) [17], meaning that the fact that there is residual disease is what really matters and not its magnitude. 

Important as it is, in patients undergoing surgical treatment, the R classification does not fully reflect the completeness of the resection. The R descriptors are based on the status of the resection margins but tell us nothing about the intensity of the intraoperative nodal assessment. For example, two resection specimens with negative margins, one with 18 negative removed lymph nodes and the other with just 1 negative removed lymph node, would be classified as pN0 R0. Today, we know that the number of removed lymph nodes is relevant to prognosis, but the original R classification misses this point. 

## 3. Previous Definitions of Complete Resection 

Because of the intrinsic limitations of the R classification in defining the completeness of resection, several authors and institutions added other parameters to the basic examination of the resection margins to define complete resection. Table 2 shows the basic elements of four different definitions of complete resection [9,19,20,21].

The Japanese definition was very strict. In addition to the integrity of the suture line, which is a resection margin, and the lymphadenectomy, which is included in all other definitions, the involvement of the visceral pleura and of the mediastinal lymph nodes, even if they had been removed, were considered features that did not qualify to define the resection as complete [9]. Mountain introduced the surgeon’s own responsibility in removing the whole lesion, the concept of the most distant node and the integrity of the nodal capsule. The involvement of the most distant node from the primary tumor and the nodal capsule were considered criteria for incomplete resection [19]. Martini and Ginsberg considered that a complete resection could only be achieved by lobectomy or pneumonectomy and emphasized the importance of avoiding transecting the tumor during dissection and of resecting invaded adjacent structures en bloc [20]. Finally, the Bronchogenic Carcinoma Cooperative Group of the Spanish Society of Pneumology and Thoracic Surgery took the most relevant criteria from the previous definitions to be applied in all resections performed by the members of the group [21]. 

## 4. The IASLC Staging Project 

In 1996, when the sixth edition of the TNM classification already was in press and about to be released, the International Workshop on Intrathoracic Staging took place in London, United Kingdom, under the auspices of the IASLC. Organized by Peter Goldstraw at the Royal Brompton Hospital, the meeting summoned speakers from North America, Japan and Europe, who lectured on many aspects of lung cancer staging [22]. The forthcoming sixth edition of the TNM classification raised critical discussions among the participants. They praised Clifton Mountain’s effort in collecting and analyzing a database of 5319 patients that had been used to inform the second to the sixth editions of the TNM classification of lung cancer, but his database had important limitations. The patients originated from a single geographic region—North America—and had undergone surgical resection of their lung cancers with no representation of other therapeutic modalities. It was evident that to make any further progress in subsequent revisions of the classification a different database was needed: one truly international and with patients treated with more varied therapies. 

The Workshop concluded by establishing three main objectives: (1) the creation of a Staging Committee, later renamed Staging and Prognostic Factors Committee (SPFC), with the responsibility to collect an international database of lung cancer patients treated with all types of therapeutic modalities to inform future editions of the TNM classification of lung cancer; (2) the formulation of an international and multidisciplinary definition of complete resection; and (3) the design of a pulmonary and mediastinal lymph node map to reconcile the differences between the Naruke and the Mountain and Dresler maps that could be used internationally. Twenty-five years after the 1996 London Workshop, all these objectives have been achieved [23]. The databases collected by the SPFC were used to inform the seventh and the eighth editions of the TNM classification of lung cancer [24,25], and a new database is being collected at the time of this writing for the revision of the eighth edition towards the ninth, due to be published in 2024. The IASLC definition of complete resection was published in 2005 [26], and the lymph node map proposed by the IASLC, in 2009 [12]. 

## 5. The IASLC Definitions of Completeness of Lung Resection

The definitions described above were used in their respective countries and working groups, but none achieved international implementation. The IASLC recognized this and created a commission to formulate a definition that could be used internationally to fulfill one of the objectives of the IASLC Staging Project. The previous definitions, the different forms of intraoperative nodal assessment and the prognostic impact of pleural lavage cytology, among other issues, were thoroughly reviewed and discussed in the SPFC. The results of those discussions were the following definitions [26]:

Complete resection must fulfill all of the following: -The resection margins (bronchial, vascular, peribronchial, around the tumor or the margins of any resected tissue) must be free of tumor proved microscopically. -The lung resection has to be accompanied by a systematic nodal dissection or by a lobe-specific systematic nodal dissection. Table 3 shows the details of the required intraoperative nodal assessment [22,26]. The minimum number of removed lymph nodes was considered to be, at least, six: three from the intrapulmonary and/or hilar nodal stations and three from the mediastinal nodal stations, always including the subcarinal. -The capsule of those nodes removed separately and of those located at the margin of the main lung specimen must be intact, without extracapsular tumor invasion.-The highest mediastinal lymph node removed must be free of tumor. 

This definition includes the R0 category, but it adds a standardized intraoperative lymph node assessment and even establishes a minimal number of resected lymph nodes.

Incomplete resection is defined by any of the following circumstances:-Tumor invasion of resection margins.-Extracapsular involvement of lymph nodes excised separately or of those at the margin of the lung specimen.-There is evidence of involved lymph nodes, but they have not been removed.-There is positive pleural or pericardial effusion. 

Incomplete resection is the equivalent of the R1 and R2 categories. The rationale for including extracapsular invasion as a criterion of incomplete resection was that, even if the involved lymph nodes are completely removed, cancer cells might be left behind in the remaining tissue surrounding the lymph nodes. It is a well-known fact that the extracapsular nodal invasion confers a poor prognosis when compared with nodal involvement without invasion of the nodal capsule. Extracapsular nodal invasion is a frequent finding. Nomura et al. reported that 29 (36%) of 80 pN1 and 57 (65%) of 88 pN2 carcinomas had involvement of the nodal capsule. Overall and recurrence-free survival were significantly worse for patients with pN1 and pN2 with extracapsular nodal invasion when compared with their counterparts without involvement of the nodal capsule [27]. Lee et al. found that the 5-year survival rate of patients with stage IIIA carcinomas without extranodal invasion was significantly better than that of patients with stage II cancers with extranodal involvement: 30.4% and 16.8% (*p* = 0.001), respectively [28]. A recent meta-analysis of 13 studies including 1709 patients (573 with and 1136 without extracapsular invasion) showed significantly increased risk of mortality from all causes in those with extranodal invasion (relative risk (RR) = 1.39, *p* < 0.0001) as well as a significantly increased risk of disease recurrence (RR = 1.32, *p* = 0.02) [29]. It has also been suggested that patients with pN2 carcinomas with extracapsular invasion do not benefit from postoperative radiotherapy [30]. All these data confirm that it is reasonable to consider extracapsular lymph node invasion as a criterion of incomplete resection. Its presence should be recorded in the pathology report [30] and should be considered in the TNM classification because it can be useful to stratify postoperative outcomes and future clinical trials [31]. Similarly, if cytological examination of pleural and pericardial effusions indicates the presence of cancer cells, even in the absence of visible tumor implants, the likelihood of remaining cancer cells in the pleural or pericardial spaces is very high. 

Once these definitions were formulated, a question was raised concerning those resections in which there was no evidence of residual disease, but all the criteria of complete resection were not fulfilled. The SPFC decided to call these resections uncertain.

Uncertain resections must have all margins free of tumor, but are associated with one or more of the following situations: -The intraoperative lymph node assessment does not achieve the standards of systematic nodal dissection or lobe-specific systematic nodal dissection.-The highest mediastinal lymph node removed is involved.-There is carcinoma in situ at the bronchial margin.-Pleural lavage cytology is positive. 

The involvement of the highest mediastinal node removed was considered an uncertain resection because it cannot be assured that the next node is negative. It could be positive, in which case the resection would be incomplete, or it could be negative, in which case the resection would be complete, provided that it fulfilled all the other criteria of complete resection. In many cases, the presence of carcinoma in situ at the bronchial margin is unrelated to the resected tumor, that is, they are different tumors. In addition, carcinoma in situ does not always progress to invasive carcinoma or has an adverse impact on prognosis [32]. This is the main reason why this situation was included in the uncertain resection. A positive cytological examination of the pleural lavage was considered uncertain because, although it generally confers worse prognosis than that assigned by the TNM classification of the tumor, not all lung cancers with positive pleural lavage cytology recur. Some studies suggest that when pleural lavage cytology is positive, the tumor should be classified as stage IIIB or, at least, it should be upstaged to the next higher T category [33,34]. 

The uncertain resection had no equivalent in the original R classification, but it was accepted as a new R category in the seventh edition of the TNM classification published by the UICC. It was coded as R0(un) and defined for all malignancies as a resection with no macroscopic or microscopic evidence of residual disease, but in which nodal assessment is based on less than the number of lymph nodes/stations ordinarily included in a lymphadenectomy specimen (3 hilar/intrapulmonary and 3 mediastinal, always including the subcarinal, for lung cancer resections), and, specifically for lung cancer, the highest mediastinal lymph node removed/sampled is positive [35]. Carcinoma in situ at the bronchial resection margin and positive pleural lavage cytology were excluded from the R0(un) category by the UICC because they already had been coded as incomplete resections: R1(is) [36] and R1(cy+) [35], respectively. 

## 6. Validation of the IASLC Definitions 

Four different validations of the IASLC definitions have been published in the past 4 years. Table 4 shows the relevant data of these studies [17,37,38,39]. 

The strength of these validations resides in the fact that statistically significant differences in prognosis were found among the three types of resection (complete, uncertain and incomplete) in four databases from four different countries of three continents in patients who had been operated on for lung cancer over a period of two decades, and that these differences were observed in resected tumors with and without nodal involvement. 

There are other interesting findings. In the series of Gagliasso et al., in the group of incomplete resections, pT4 and pN2 tumors had the worst prognosis, and exploratory thoracotomies had a worse prognosis than incomplete resections: 5-year survival rates of 10.5% and 15.7%, respectively (*p* = 0.0001). Among patients with uncertain resections, the worst prognosis was for those with positive highest mediastinal node, with a 5-year survival rate of 28.8%, while the 5-year survival rates were much higher for those with suboptimal nodal assessment and those with carcinoma in situ at the bronchial margin: 44.2% and 40%, respectively [37]. 

In the series of the IASLC, Edwards et al. found that the differences among the three types of resection were maintained in patients with tumors free from nodal disease and in those harboring nodal involvement. There were no statistically significant differences in survival of patients with R1 and R2 resections. In those with pN2R0, the involvement of the highest mediastinal lymph node removed had a worse prognosis compared with that of patients without this type of involvement. However, this difference was not found in the group of patients with pN2R1 tumors. In addition, no significant differences were found among patients with pN1 and pN2 tumors with or without extracapsular nodal invasion [17]. 

In the population-based series of Osarogiagbon et al., similarly to the finding in the IASLC database, the differences among the three types of resection were also found in the population of patients with tumors with and without nodal disease. Among patients with R0 resections by UICC criteria, 64% had uncertain resections mainly because of suboptimal intraoperative nodal assessment. In those patients with uncertain resection, those with positive highest mediastinal node had a worse prognosis than those with suboptimal lymphadenectomy. All incomplete resection criteria were associated with a similar prognosis, but all had a worse prognosis than complete and uncertain resections. An important original finding in this series was the fact that there was a progressive degradation of survival depending on the thoroughness of the intraoperative nodal assessment. The median survival for complete resection, uncertain resection with at least one mediastinal lymph node sampled, uncertain resection with no mediastinal lymph nodes sampled, uncertain resection with pNX and incomplete resection were: not reached, 77, 65, 50 and 25 months, respectively. Their adjusted hazard ratios were 1, 1.28, 1.47, 1.74 and 2.18, respectively (*p* < 0.0001) [38]. 

In the specific series of patients with pN2 tumors, Yun et al. found statistically significant differences in overall and progression-free survival in patients with R0 and R1–R2 resections by UICC criteria, and in those with complete, uncertain and incomplete resections. However, these differences in the three types of resection were not found when pN2 tumors were subdivided according to the IASLC proposal into pN2a1 (involvement of a single N2 station without N1 disease), pN2a2 (involvement of a single N2 station with N1 disease) and pN2b (involvement of multiple N stations). In univariate and multivariate analyses, the IASLC type of resection was a significant prognostic factor, although the significance between complete and uncertain resections was lost after adjustment of covariates [39].

## 7. Requirements for a Complete Resection 

Complete resection is performed in the operating room, but it requires days and perhaps weeks of planning before and after the operation. A multidisciplinary team must assess the patient. The tumor must be adequately staged. The resection must follow certain rules to provide therapeutic value to the patient. The pathologic study of the specimen must be as thorough as possible. 

### 7.1. Multidisciplinary Team Assessment

There is no doubt today that the assessment of each patient with lung cancer by a multidisciplinary team of specialists has prognostic value. In the study of Osarogiagbon et al., the outcomes of 376 patients seen at the University of Tennessee Cancer Institute, in the United States of America, were analyzed according to whether the recommendations of the multidisciplinary team were followed or not. The management of 236 (63%) patients was in accordance with the recommendations and that of 139 (37%) was not. For those with concordant management, the onset of definitive therapy was shorter (14 days vs. 25 days, *p* < 0,002), and the median overall and progression-free survival were longer (2.1 years vs. 1.3 years, *p* = 0.001; and 1.3 years vs. 0.8 years, *p* < 0.05, respectively) when compared with those of the patients whose management did not follow the multidisciplinary team recommendations. Issues related to the patients’ insurance policy and to the managing physician’s decisions were the main reasons for discordant management [40]. Tamburini et al., in Italy, in a propensity score-matched study, selected 170 patients from 246 that were surgically treated from 2008 and 2012 and who had not undergone multidisciplinary evaluation, as well as 170 from 231 that were operated on from 2012 and 2015 and who had been evaluated by a multidisciplinary team. More patients seen by the multidisciplinary team had complete preoperative evaluation: 159 (93%) vs. 109 (64%), *p* < 0.001. Stage III lung cancers were more frequent in the group that was not seen by the multidisciplinary team: 41 (24%) vs. 26 (15%), *p* = 0.041. One-year mortality was higher in the group that was not evaluated by the team: 18% vs. 8%, *p* = 0.006. Moreover, 1-year survival was better for those patients evaluated in a multidisciplinary discussion: odds ratio 0.48 (95% confidence interval 0.25–0.92) [41]. A recent study from Taiwan by Hung et al., focusing on 515 patients with stage III lung cancers, showed that those treated after a multidisciplinary team discussion had a significantly longer median survival time when compared with those managed without multidisciplinary evaluation: 41.2 months vs. 25.7 months, *p* = 0.018. Multidisciplinary team evaluation remained a significant prognostic factor in the multivariate analysis together with T category, performance status and surgical resection [42]. These reports highlight the relevance of the multidisciplinary team in the management of patients with lung cancer. This relevance can be replicated in different countries by different medical teams. 

### 7.2. Strict Tumor Staging

In addition to a preoperative cardio-respiratory assessment to evaluate the patient’s capacity to undergo the planned resection and to ensure an acceptable postoperative quality of life [43], the tumor must be adequately staged. This means that the staging procedure must follow the evidence-based guidelines available today. The guidelines of the American College of Chest Physicians (ACCP) are comprehensive and well documented with the best available evidence [44]. In short, all patients who are deemed candidates for lung cancer resection should have a positron emission tomography–computed tomography (PET-CT). For patients with ground glass nodules suggestive of adenocarcinoma in situ or minimally invasive adenocarcinoma with no other lung abnormality, PET is not required. Other tests will be ordered depending on the findings of these basic explorations, and should be indicated sequentially and with an increasing degree of invasiveness [45]. 

The guidelines of the European Society of Thoracic Surgeons (ESTS) focus on preoperative mediastinal staging [46], and their recommendation on invasive staging is in agreement with the recommendation of the ACCP. In essence, invasive mediastinal staging is indicated when mediastinal lymph nodes are abnormal, either by anatomic or metabolic criteria or both, or when the probability of mediastinal nodal disease is high, such as in central tumors or when there is evidence of hilar nodal involvement. Both societies agree that, if available, ultrasound-assisted endoscopic procedures (endobronchial ultrasound–transbronchial needle aspiration, endoscopic ultrasound–fine-needle aspiration or their combination) should be indicated first, but if their results are negative, a minimally invasive surgical procedure should be indicated for confirmation [44,46]. Video-assisted mediastinoscopy is the most commonly used confirmatory procedure, but video-assisted thoracoscopy or parasternal mediastinotomy can also be indicated depending on the location of the suspicious lymph nodes. The accuracy of these procedures depends on how the procedure is performed and on the thoroughness of the exploration. These minimally invasive surgical procedures must be performed to their maximum capacity to take the greatest advantage of the operation [47]. 

In the absence of metastatic disease, the status of the regional lymph nodes leads prognosis. Failure to properly stage tumor extent to the lymph nodes increases the probability of underappreciated and ignored nodal disease that may lead to incomplete resection, multilevel nodal involvement or involvement of the subcarinal lymph nodes, which are associated with much lower survival than that expected from surgical treatment [48]. 

### 7.3. Correct Surgical Technique and Intraoperative Staging

The process to achieve a complete resection is associated with the determination of the pathologic classification of the tumor. Rule number one of the general rules of the TNM classification states that the tumor must be resected or, if this is not possible, an adequate biopsy should be taken to define the highest T category [16], in which case, the resection will be incomplete. One of the definitions of complete resection requires that the tumor is not transgressed during its resection, and that the invaded adjacent structures are removed en bloc to avoid dissemination of cancer cells into the operative field [20]. As for the pathologic assessment of the regional lymph nodes, the removal of the lymph nodes must be adequate to determine the absence of nodal disease or the highest N category [16]. As mentioned in Section 5 and Table 3, this is best done by systematic nodal dissection or lobe-specific systematic nodal dissection. The former is preferable because, in a recent meta-analysis, it has shown a significant advantage in long-term survival [49]. For lung cancer, one of the conditions of complete resection is that the lymphadenectomy specimen must contain, at least, three mediastinal lymph nodes (one always from the subcarinal station) and three from the hilar and intrapulmonary nodal stations, otherwise the resection would be considered uncertain [35]. In stage IV carcinomas deemed resectable, both the primary and the metastatic sites must be removed. The margins of the metastatic specimen must be free of tumor to consider the resection as complete. 

### 7.4. Complete Pathologic Study and Staging

The pathologic study of the resected specimens must provide the final diagnosis of cell type and differentiation grade, the size of the tumor and the status of the removed adjacent structures. Lymphatic permeation, vascular invasion and perineural invasion are optional descriptors of the TNM classification, and should be examined and included in the final pathology report. All of them have prognostic relevance and can qualify the prognosis given by the TNM of the tumor [50,51,52]. All mediastinal, hilar and interlobar lymph nodes removed and labeled by the surgeon should be studied for tumor invasion including the status of the nodal capsule. As discussed in Section 5, this feature of invasiveness is associated with a worse prognosis and is a criterion of incomplete resection. The pathologist will examine the resected lung and will search for intrapulmonary nodes as thoroughly as possible. The intensity of this examination has staging and prognostic importance. As shown by Ramirez et al., when the histopathological study is intensified beyond standard practice, more intrapulmonary nodes are found, and 12% of them are involved by cancer [53]. This means that, from the standard pathologic study, these tumors would have been classified as pN0, when they were, in fact, pN1. When the newly found nodes are involved, prognosis is significantly worse, especially when there are more than two involved nodes [54]. The therapeutic consequences of this are obvious: the patients with undetected pN1 nodes would have not been considered for adjuvant chemotherapy and would have not benefited from the added survival, and the prognosis assigned to their disease based on the TNM classification would have been wrong. All removed and involved lymph nodes must be counted because quantifying nodal disease in lung cancer has prognostic implications. In the seventh and eighth editions of the TNM classification of lung cancer, it was found that the number of involved nodal zones and nodal stations separated groups of tumors with statistically significant survival differences [55,56]. 

## 8. Potential Future Refinements

The TNM classification determines the anatomic extent of the tumor supported by cyto-histological confirmation. In the TNM classification of lung cancer, there are cancer cells in different anatomic areas that have their own code in the classification (Table 5) [16]. 

These cancer cells define certain T, N and M categories or define the type of resection as incomplete. Two relatively new situations that have important prognostic implications do not have, yet, any code in the TNM classification: cancer cells spread through air spaces (STAS) and circulating tumor cells or their components, namely, circulating tumor DNA and RNA. 

In 2015, Kadota et al. triggered the alarm regarding the presence of cancer cells beyond the limits of resected lung adenocarcinomas less than 2 cm. They found STAS in 155 (38%) of 411 resected adenocarcinomas. Those patients who had undergone sublobar resection and whose tumors had STAS had a significantly higher rate of recurrence, but this difference was not observed in those treated with lobectomy [57]. Two years later, Dai et al. described STAS in lung adenocarcinoma of more than 2 but less than 3 cm in size, and, because of the adverse effect of STAS on prognosis, suggested that T1cN0M0 (stage IA3) tumors with STAS should be classified as stage IB [58]. During the past few years, STAS has been described in squamous cell carcinoma [59], in small cell lung cancer [60,61], in pleomorphic carcinoma [62], in atypical carcinoids and large cell neuroendocrine carcinomas [61]. In every instance, STAS has been associated with poor prognosis compared with the prognosis of those cancers with no STAS. STAS is identified in the resected specimen, so it is unknown if there are cancer cells in the remaining lung. However, considering the associated poor prognosis of STAS and the possibility that more cells can be spread in the remaining lung, perhaps resections with STAS could be defined as uncertain. 

Research on liquid biopsy and its clinical applications are expanding very quickly according to the increasing number of articles published on the topic. The term now does not only include blood determinations but also includes urine, saliva, breath and feces. Circulating tumor cells, circulating DNA, microRNA, extracellular vesicles and epigenetic signatures, among others, are commonly determined in blood, urine and saliva samples, while volatile compounds and gut microbiota are determined in breath and feces samples, respectively. [63]. However, blood samples are the ones most commonly utilized. The determination of biomarkers in liquid biopsy is useful in early diagnosis of lung cancer [64,65], in monitoring patients and genetic alterations during therapy [63,66], in characterizing molecular tumor profile [66,67], in assessing prognosis [68] and in detecting minimal residual disease in early stage lung cancer after curative resection [69]. Regarding the latter, Wu et al. determined the circulating tumor cell count before and after resection on the day of surgery, and on postoperative days 1 and 3 in 41 patients with lung cancer, 30 (80%) adenocarcinomas. The count dropped on postoperative day 1, but a rise in count on day 3 in 11 patients was associated with recurrence [69]. Chaudhuri et al. determined circulating tumor DNA in 40 patients before and after intended curative treatment with radiotherapy, chemotherapy and/or surgery for stage I to III lung cancer. Circulating tumor DNA was found at least in one post-treatment determination in 20 (54%) of 37 patients who had had pre-treatment circulating tumor DNA. All of these 20 patients experienced tumor recurrence and had significantly lower freedom from progression and lower survival than those without circulating tumor DNA (*p* < 0.001). The presence of circulating tumor DNA preceded radiographic progression in 72% of these patients by a median of 5.2 months, and 53% had had circulating tumor DNA mutations profiles associated with favorable responses to tyrosine kinase inhibitors and immunotherapy [70], which is a useful finding to design a more personalized treatment plan. 

Cancer cells or their fragments in the blood could be considered as the cells in other body spaces or anatomic structures (Table 5). They are in a fluid tissue, blood, in a well-defined anatomic system, the circulatory, but to date, they are not coded in the TNM classification. Yang et al. have suggested adding the letter B, for blood, to the TNM (TNMB) as a new component of the classification [71]. That would certainly complement the TNM classification, but liquid biopsy is not universally available and perhaps it is too early to consider it in the TNM classification. It also requires more research to quantify the amount of cells or of circulating tumor DNA or RNA that make a difference in prognosis. However, the presence of tumor material in the blood after treatment should be considered as minimal residual disease, although more research is needed to see how to classify it and how to manage it. 

Ginsberg and Martini did not consider sublobar resections to qualify for complete resection [20]. The IASLC definitions of completeness of resection did not touch on this point. Compared with sublobar resections in large databases, lobectomy has a significant better survival: hazard ratio (HR) 0.62, *p* = 0.0001 [72]. Sublobar resections include two different types of operations from the technical and prognostic points of view: segmentectomies and wedge resections. Compared with segmentectomies, wedge resections for stage IA tumors are associated with a significantly higher rate of resection margins less than 1 cm (62% vs. 27%, *p* = 0.003) and of local recurrence (14.5% vs. 3.8%, *p* = 0.002) [73]. Significant differences in 5-year survival rates (80% vs. 48%, *p* = 0.005) and in local recurrence rates (11% vs. 40%, *p* = 0.007) between segmentectomy and wegde resection, respectively, were found even in tumors 2 cm or less in greatest dimension [74]. A recent meta-analysis comparing segmentectomy with wedge resection has found significant differences in overall survival (HR 0.82, *p* = 0.00001), cancer-specific survival (HR 0.71, *p* = 0.00001) and disease-free survival (HR 0.73, *p* < 0.04) [75]. However, there are patients with a very long survival after sublobar resections. The associated poor prognosis may not arise from the type of resection alone, but also from inadequate margins and suboptimal intraoperative lymph node assessment, both common in sublobar resections. Regarding the value of lymphadenectomy in sublobar resections, it has been found that survival differences between lobectomy and sublobar resection disappear when the lymphadenectomy specimen includes nine lymph nodes or more [76]. The still on-going North American (Cancer and Leukemia Group B 140503) and Japanese trials (Japan Clinical Oncology Group JCOG0802 and 0804) will set the indications for wedge resection and segmentectomy for part-solid tumors and solid tumors of 2 cm or less in greatest dimension. It will then be important to define the adequate resection margins to avoid local recurrence, and to establish a standard intraoperative nodal assessment for these small tumors. 

As shown in the study of Osarogiagbon et al., 64% of R0 resections by the original R0 descriptor were reclassified as uncertain resections mainly due to an intraoperative nodal assessment that did not meet the standards of systematic nodal dissection or of lobe-specific systematic nodal dissection [38]. This is a serious matter because uncertain resections have a significantly poorer prognosis than complete resections (Table 4). In fact, suboptimal nodal dissection is quite common. In a series of 12,349 patients registered from 1998 to 2002 in the Surveillance Epidemiology and End Results (SEER) database and who had undergone lung cancer resection, 7711 (62%) had no mediastinal lymph nodes retrieved. There were no statistically significant differences between those who had mediastinal dissection and those who did not regarding 30-day mortality: 0.64% and 0.96% (*p* = 0.07), respectively. However, those who did not have mediastinal nodal dissection had a significantly higher cancer-related death rate (62% and 40%, *p* < 0.001) and lower overall 5-year survival and 5-year cancer-specific survival rates than those who had mediastinal lymph node assessment (47% and 52%, *p* < 0.001, respectively; and 58% and 63%, *p* < 0.001, respectively) [77]. The lung resections performed during the period of registration in the SEER database are over 20 years old. More contemporary data, however, show that the recommended standard for intraoperative nodal assessment is not always achieved [78,79], although nodes removed at mediastinoscopy and nodal fragmentation may be confounding factors regarding the total number of removed lymph nodes [80]. 

Sooner or later, quantification of nodal disease to assign an N category to lung cancer will be introduced in the TNM classification. The number of involved lymph nodes already is a descriptor in cancers of the gastrointestinal tract, lip, oral cavity, skin, breast, penis, testes, urinary tract and vulva [16]. If this is the case of lung cancer, an adequate lymphadenectomy will become even more important because quantification of nodal involvement and, therefore, the assignment of an N category based on the number of involved lymph nodes, will depend on its thoroughness. There are several ways to improve the quality of lymphadenectomy. Training during residency is fundamental to achieve the necessary technical skills and also to understand the oncological concept of lung cancer resection. The role of scientific societies should be emphasized. Their recommendations are based on the best evidence and deserve to be followed [81]. From the practical point of view, there are lymph node trays [82] and lymph node kits [83] that have been designed to improve the quality of intraoperative nodal staging. The use of the lymph node kit significantly increased the number of mediastinal lymph nodes examined, the percentage of resections meeting the recommendations for mediastinal nodal dissection and the proportion of patients eligible for adjuvant therapy trials (*p* < 0.001), without increasing the duration of surgery or the postoperative complications [83]. It proved useful to determine the extent of the lymphadenectomy performed, which, sometimes, is not reflected in the operative notes [84]. When adjusted for confounders, the use of the kit was associated with a nearly significant survival benefit: hazard ratio 0.85 (95% confidence interval: 0.71-1.02, *p* = 0.0716). Moreover, the kit increased the number of patients in whom at least three mediastinal nodal stations had been examined, and these patients had a significantly better survival than those whose operation did not meet that standard (*p* = 0.0125) [85]. At the time of this writing, the N Subcommittee and the Lymph Node Map Subcommittee of the IASLC Staging and Prognostic Factors Committee are discussing ways to improve the nodal chart, facilitate the identification of the different nodal stations and improve nodal dissection by creating more realistic images and providing videos showing the limits of the nodal station and their dissection [86]. 

## 9. Conclusions

The concept of complete resection has been evolving during the past four decades, from the relatively simple residual tumor descriptor to the more elaborated definitions of the IASLC. These definitions should not be considered the last step in the progress of defining complete resection. There are new issues that should be integrated into future refinements of the definition, such as STAS, tumor cells or their DNA and RNA identified in the patients’ blood, and the minimal margins and adequate lymphadenectomy for sublobar resections. The definitions and their requirements clearly show that to achieve a complete resection is not only a surgical gesture but a truly multidisciplinary activity of many specialists involved in the patient’s evaluation, tumor staging and pathologic examination of the resected specimens. 

The IASLC definitions of complete, uncertain and incomplete resections include the basic components of previous definitions and have been widely validated in different databases. These validations have proved the significant prognostic value of the three types of resection, and, therefore, they should be taken into account when planning and performing a resection for lung cancer. 

## Figures and Tables

**Table 1 cancers-13-02583-t001:** Residual tumor classification.

Categories	Descriptors
RX	The presence of residual tumor cannot be assessed
R0	There is no residual tumor
R1	There is microscopic residual tumor
R2	There is macroscopic residual tumor

**Table 2 cancers-13-02583-t002:** Elements used to define complete resection.

Naruke et al. 1978 [9]	Mountain 1983 [19]	Martini and Ginsberg 1995 [20]	Bronchogenic Carcinoma Cooperative Group 1998 [21]
Visceral pleura	Surgeon’s assessment	Lobectomy or pneumonectomy	Resection margins
Suture line	Most distant lymph node	Tumor integrity and en bloc resection	Most distant lymph node
Mediastinal lymph nodes	Resection margins	Resection margins	Lymph node capsule
Complete lymphadenectomy	Lymph node capsule	Mediastinal lymphadenectomy	Mediastinal lymphadenectomy

**Table 3 cancers-13-02583-t003:** Required intraoperative nodal assessment for complete resection.

Type of Lymphadenectomy	Requirement *
Systematic nodal dissection	Step (1) Complete excision of the mediastinal fat and enclosed lymph nodes, which are dissected and identified in accordance with an internationally accepted nodal chart. Step (2) Excision of hilar and intrapulmonary lymph nodes and their identification in accordance with an internationally accepted nodal chart. Dissection should proceed in a centrifugal manner until the extent of resection has been determined.
Lobe-specific systematic nodal dissection	For right upper and middle lobes: subcarinal, superior and inferior paratracheal lymph nodes. For right lower lobe: subcarinal, right inferior paratracheal and either the paraesophageal or pulmonary ligament lymph nodes. For left upper lobe: subcarinal, subaortic and para-aortic lymph nodes. For left lower lobe: subcarinal, paraesophageal and pulmonary ligament lymph nodes. For all lobes: dissection and histological examination of hilar and intrapulmonary (lobar, interlobar, segmental) lymph nodes.

* The recommended nodal chart is the one proposed by the IASLC. The nodal stations mentioned in this table are in accordance with this nodal chart [12].

**Table 4 cancers-13-02583-t004:** Characteristics and survival rates of the published validations of the IASLC definitions of completeness of resection.

Characteristics	Gagliasso et al. [37]	Edwards et al. [17]		Osarogiagbon et al. [38]	Yun et al. [39]
Year	2017	2019		2019	2021
Type of study	Single institution	International database		Population-based	Single institution
Study period	1998–2007	1999–2010		2009–2019	2004–2018
Country	Italy	World (mainly Japan)		USA	South Korea
No. of patients	1277	14,712		3361	1039
No. (%) of R0	1003 (78.5%)	6070 (41%)		1119 (33%)	432 (41.6%)
No. (%) of R0(un)	185 (14.5%)	8185 (56%)		2044 (61%)	212 (20.4%)
No. (%) of R1 + 2	89 (7%)	457 (3%) (301 + 156)		196 (6%)	395 (38%)
**5-year survival rates**					
	All pN	pN0	pN+	All pN	pN2
R0	58.8%	82%	55%	64%	54.7%
R0(un)	37.3%	79%	45%	54%	45.8%
R1 + 2	15.7%			33%	36.2%
R1		46%	34%		
R2		38%	22%		
*p* value	0.0001	0.04	<0.001	<0.0001	0.043 (R0 vs. R0(un)) 0.010 (R0(un) vs. R1 + 2)

**Table 5 cancers-13-02583-t005:** Lung cancer cells that count for the TNM classification and have their specific code.

Location of Cancer Cells	TNM Code
Bronchial secretions	T0
Pleural fluid	M1a
Pericardial fluid	M1a
Lymph nodes	N0 (i/mol+) *
Lymphatic vessels	L1
Bone marrow	M0 (i/mol+) *
Pleural lavage cytology	R1 (cy+) **

* Cancer cells identified by morphological or non-morphological analyses. ** Incomplete resection because of positive pleural lavage cytological examination.

## Data Availability

Not applicable.

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
