# Peer review of "The Evolving Concept of Complete Resection in Lung Cancer Surgery"

_cancers, 2021, doi:10.3390/cancers13112583_

Round 1
Reviewer 1 Report
Thank you for giving an opportunity for reviewing the article.
I have read this article with great interest. The manuscript is well-organized, concise and timely. This article gives readers big pictures of how the concepts of complete surgical resection for lung cancer has been established, and what is coming as the next stage. I don't think the manuscript needs major revisions.
Minor: Table 2 was a bit difficult to understand at a glance. For the sake of clarity, author may want to add columns/rows to allow the reader to see the elements side by side between definitions.
Author Response
Thank you for giving an opportunity for reviewing the article.
I have read this article with great interest. The manuscript is well-organized, concise and timely. This article gives readers big pictures of how the concepts of complete surgical resection for lung cancer have been established, and what is coming as the next stage. I don't think the manuscript needs major revisions.
RESPONSE: I highly appreciate your encouraging comments. Thank you very much.
Minor: Table 2 was a bit difficult to understand at a glance. For the sake of clarity, author may want to add columns/rows to allow the reader to see the elements side by side between definitions.
RESPONSE: Thank you for pointing out this. I have added vertical lines to the table to separate the columns. However, I think this is not the journal style, and, maybe, the Style Editor of Cancers will have the last word on this modification.
Reviewer 2 Report
Surgical resection is the most important curative treatment for lung cancer. With the successful implementation of lung cancer screening programs, the proportion of patients who undergo surgery is likely to increase significantly. The IASLC has proposed a revision of the residual disease (R-factor) classification, from complete (R0), microscopic incomplete (R1) and grossly incomplete (R2) to R0, ‘R-uncertain’, R1 and R2. The adverse prognostic impact of R-uncertainty has been independently validated, with the majority caused by poor nodal evaluation. In this perspective article, prof. Ramón Rami-Porta described different definitions that came from four validations about the prognostic relevance of surgical resection. And these definitions should be followed once you have established the surgical resection of lung cancer. In the concluding remarks, spread through air spaces, residual tumor cells, DNA/RNA in the blood, determination of the adequate margins and lymphadenectomy in sublobar resections could play a key role to define this kind of strategy.
To my eyes, the paper is an excellent guide looking at the historical background described. Moreover, this article helps to reconstruct the evolution of the concept of complete resection in the surgical treatment of lung cancer. Detailed comments of a single study surely fit into the standard quality of our journal and confirm the importance of data analysis and results discussed. The scientific content is good and the English style and language used in the manuscript is fine. Moreover, I appreciate the scientific efforts to organize this paper and I think that the concept of complete resection is well-argued. The resolution of the table of each section and the reference list meets the quality requirements of our journal. Moreover, the results were clearly discussed and corroborated with what is shown. Finally, the data analyses were interpreted in a comprehensible manner. I didn’t observe any remarkable incongruences throughout the text.
I have only two questions before considering the paper accepted:
- It is well known as poor surgical quality may reduce the survival benefit of curative-intent surgery. In addition, a suboptimal pathologic nodal evaluation is the most prevalent NSCLC surgery quality deficit. We are faced with a global problem that becomes prevalent across institutions of different characteristics. What’s the solution? Any suggestions to implement in the final section?
- The resection is based on different principles in order to gain a therapeutic value for the patient. So, I would like to extend the question about the significance of circulating tumor cells or circulating tumor DNA or RNA detected by liquid biopsy and the correlation with recurrence and disease specific survival. Please discuss it in more detail.
Author Response
Surgical resection is the most important curative treatment for lung cancer. With the successful implementation of lung cancer screening programs, the proportion of patients who undergo surgery is likely to increase significantly. The IASLC has proposed a revision of the residual disease (R-factor) classification, from complete (R0), microscopic incomplete (R1) and grossly incomplete (R2) to R0, ‘R-uncertain’, R1 and R2. The adverse prognostic impact of R-uncertainty has been independently validated, with the majority caused by poor nodal evaluation. In this perspective article, prof. Ramón Rami-Porta described different definitions that came from four validations about the prognostic relevance of surgical resection. And these definitions should be followed once you have established the surgical resection of lung cancer. In the concluding remarks, spread through air spaces, residual tumor cells, DNA/RNA in the blood, determination of the adequate margins and lymphadenectomy in sublobar resections could play a key role to define this kind of strategy.
To my eyes, the paper is an excellent guide looking at the historical background described. Moreover, this article helps to reconstruct the evolution of the concept of complete resection in the surgical treatment of lung cancer. Detailed comments of a single study surely fit into the standard quality of our journal and confirm the importance of data analysis and results discussed. The scientific content is good and the English style and language used in the manuscript is fine. Moreover, I appreciate the scientific efforts to organize this paper and I think that the concept of complete resection is well-argued. The resolution of the table of each section and the reference list meets the quality requirements of our journal. Moreover, the results were clearly discussed and corroborated with what is shown. Finally, the data analyses were interpreted in a comprehensible manner. I didn’t observe any remarkable incongruences throughout the text.
RESPONSE: Thank you very much for your detailed comments. I am very satisfied to read that the manuscript meets the standards of Cancers. I am very grateful, indeed.
I have only two questions before considering the paper accepted:
- It is well known as poor surgical quality may reduce the survival benefit of curative-intent surgery. In addition, a suboptimal pathologic nodal evaluation is the most prevalent NSCLC surgery quality deficit. We are faced with a global problem that becomes prevalent across institutions of different characteristics. What’s the solution? Any suggestions to implement in the final section?
RESPONSE: Thank you for raising the relevance of this important issue. Following your very pertinent comment, I have added two paragraphs to section 8. Potential future refinements. The first one deals with the consequences of not performing a proper intraoperative nodal assessment; and the second elaborates on different ways to improve nodal dissection at the time of tumor resection. Both are supported by new references.
- The resection is based on different principles in order to gain a therapeutic value for the patient. So, I would like to extend the question about the significance of circulating tumor cells or circulating tumor DNA or RNA detected by liquid biopsy and the correlation with recurrence and disease specific survival. Please discuss it in more detail.
RESPONSE: Thank you for highlighting this important aspect of persistent disease. I have no personal experience in liquid biopsy, but the published evidence shows that it is clinically useful for early diagnosis of lung cancer, to monitor therapies, to determine molecular tumor profile to select targeted therapies and immunotherapy, to evidence recurrence before it is clinically evident, and to indicate further therapies after the initial ones. Persistent tumor material in blood now falls into the concept of minor residual disease, which is important to monitor the effect of treatment and may be the basis to indicate additional treatments. There are important contributions on this topic in recently published articles, including some in Cancers. Therefore, I have expanded my comments in the final section of the manuscript. I have added a new paragraph and more references that will be useful to the reader.